

# Peanut oil is more environmentally sustainable than rapeseed oil from a carbon and nitrogen footprint perspective in China

Fen Ma[1], Mingbao He[2], Yingchun Li[2], Yanqun Wang[3], Zhengping Peng[3,4], Yinlong Xu[2], Bohan Zhao[2] and Jingyu Zhang[2]

[1] College of Digital Economy, Fujian Agriculture and Forestry University, Fuzhou, China
[2] Institute of Environment and Sustainable Development in Agriculture, Chinese Academy of Agricultural Sciences, Beijing, China
[3] College of Resources and Environmental Sciences/Key Laboratory of Farmland Eco-Environment of Hebei, Hebei Agricultural University, Baoding, China
[4] State Key Laboratory of North China Crop Improvement and Regulation, Baoding, China

Corresponding author
Yingchun Li, liyingchun@caas.cn

## ABSTRACT

Peanut and rapeseed oil, prominent edible oils in China, significantly contribute to greenhouse gas and reactive nitrogen emissions. A comprehensive examination of their environmental footprints is foundational for developing green and low-carbon products. Using a cradle-to-factory gate life cycle assessment, we quantified the carbon footprint (CF) and nitrogen footprint (NF) associated with the oil production of peanut and rapeseed from 2004 to 2023 in China. The results showed that peanut oil has a lower environmental impact than rapeseed oil, with a CF of 3,312.2 kg $CO_2$eq $t^{-1}$ oil and NF of 28.5 kg reactive nitrogen (Nr) $t^{-1}$ oil, respectively, compared to 3,722.4 kg $CO_2$eq $t^{-1}$ oil and 43.3 kg Nr $t^{-1}$ oil for rapeseed oil. It corresponded to less than 11.0% in CF and 34.2% in NF of peanut oil than that of rapeseed oil. The cropping phase was the primary source of disparity between the two oil products, with peanut exhibiting consistently lower yield-based CF and NF than rapeseed. Fertilizer application, primarily nitrogen (N) and compound fertilizers, accounted for 63.7% (peanut) and 91.4% (rapeseed) of CF, meanwhile N runoff and ammonia ($NH_3$) volatilization were dominant in NF. Moreover, regions such as Jiangxi (peanut) and Yunnan, Shaanxi, and Gansu (rapeseed) exhibited high CF and NF but low productivity, suggesting the need for cropping layout optimization. Our findings highlight the environmental advantages of peanut oil, and recommend improved fertilizer management in agricultural stage and cleaner oil processing production to promote low-carbon, sustainable edible oil production in China.

## INTRODUCTION

The massive emissions of carbon and nitrogen from crop production, including greenhouse gas (GHG) emissions (*Tubiello et al., 2013*) and reactive nitrogen (Nr) losses (*Galloway et*

*al., 2008*), are driving global climate warming and environmental deterioration. With an increasing momentum for worldwide carbon mitigation, green low-carbon products will become a preferred consumption choice (*Li, Long & Chen, 2017*). Edible oil production is a considerable source of GHG emissions and Nr losses (*Alcock et al., 2022*), making low emission edible oils more competitive. Peanut (*Arachis hypogaea* L.) and rapeseed (*Brassica napus* L.) are the most important oil-bearing crops concerning the supply of edible oil, grown in more than 100 countries and 66 countries in the world, respectively (*FAO et al., 2022*). China is a major grower of peanut and rapeseed worldwide, with 4.68 million hectares of peanut, second only to India, and 7.25 million hectares of rapeseed, ranking third in 2022 (*National Bureau of Statistics of China, 2023*). Peanut and rapeseed oil, the major edible oils in China, accounted for approximately 77.9% of the domestic edible vegetable oil production (*He et al., 2022*; *Liao, 2020*). Among domestic edible oils (domestic raw materials), the consumption of peanut oil accounted for about 23% while that of rapeseed oil accounted for approximately 37% in 2022 (https://www.lswz.gov.cn/html/zmhd/lysj/index.shtml). They also faced the challenges of insufficient domestic supply of oil-bearing seeds and a low edible oil self-sufficiency rate. To ensure the safety of edible oil, the Chinese government encourages the expansion of the cultivation area and edible oil production of oil-bearing crops in an environmentally sustainable way. Therefore, evaluating the environmental impacts of edible peanut and rapeseed oils is imperative, providing scientific insights for industry, consumers, and policymakers in promoting green, low-carbon products.

In order to advance the development of eco-friendly agricultural products, comprehensively accounting for total GHG emissions and Nr losses within a product system is of significant importance. The term carbon footprint (CF) is defined as the total amount of GHG emissions directly or indirectly caused by anthropogenic activities throughout the life stages of a product or a service expressed in carbon dioxide equivalent ($CO_2$eq) (*Gan et al., 2011*). Similarly, the nitrogen footprint (NF) has been defined as the total amount of Nr released into the environment during resource utilization activities (*Leach et al., 2012*). In recent years, the process-based life cycle assessment (LCA) method has been widely used in calculating the CF or NF of agricultural systems, including peanut and rapeseed (*Chen et al., 2021*; *Hosseinzadeh-Bandbafha et al., 2018*; *Li et al., 2022*; *Macwilliam et al., 2016*). This method can provide valuable reference information for reducing CF or NF in crop planting. However, limited studies have evaluated the CF and NF of edible rapeseed oil or peanut oil products, with most focusing primarily on the CF (*Badey et al., 2013*; *Bai et al., 2021*; *He et al., 2021*; *Ji et al., 2021*; *Schmidt, 2015*). The majority of studies on the environmental impacts of rapeseed oil production have primarily focused on its application in biofuel production (*Shi et al., 2017*; *Uusitalo et al., 2014*). When evaluating the CF of edible rapeseed oil in European countries, the study of *Schmidt (2015)* incorporated carbon emissions from land use changes, while the results of *Badey et al. (2013)* did not consider emissions from the cropping system. In China, *Bai et al. (2021)* compared the environmental performance, including the CF of edible soybean oil, peanut oil, and rapeseed oil, using the LCA approach, revealing better environmental performance for rapeseed oil and peanut oil compared to soybean oil. However, this study

did not quantitatively assess the NF of edible peanut and rapeseed oil, despite the Nr losses incurred during agricultural inputs such as fertilizer, agricultural film, and pesticides (*Li et al., 2022*). Compared to a single footprint indicator, an integrated assessment of multiple footprints offers a more comprehensive basis for environmental decision-making and policy recommendations. *He et al. (2021)* evaluated the CF and NF of peanut and edible peanut oil in China from 2008–2017 in China yet the NF did not include NO emissions, which contributed to 17% of all Nr losses (*Li et al., 2022*). Currently, there is no existing research that jointly analyzes the CF and NF for edible oil production of peanut and rapeseed in China. Sustainable agricultural development necessitates the reduction of carbon emissions and the simultaneous improvement of carbon and nitrogen efficiency per unit product (*Wang, Zhang & Zhang, 2019*). Therefore, conducting a comprehensive analysis of the CF and NF of edible peanut oil and rapeseed oil is a necessary step toward promoting green and low-carbon products in the future. As a large country involved in peanut and rapeseed planting and oil production, the CF and NF of peanut and rapeseed oil production in China should be systematically studied.

The primary objective of this study is to enhance the agricultural CF and NF database in China and to offer evidence-based recommendations for policymakers aimed at reducing the footprint associated with peanut and rapeseed oil production. Given the aforementioned research limitations, this study compared the comprehensive CF and NF of edible peanut and rapeseed oil production in China based on national statistical data from 2004 to 2023, using the LCA method performed from the cradle to the product factory. Specifically, the main objectives of this article are: (1) to present spatial–temporal changes in CF and NF of peanut and rapeseed planting phases; (2) to identify the low-carbon green edible oil product between peanut oil and rapeseed oil. This study offers scientific guidance for reducing the environmental impacts of edible oil production in China.

## MATERIALS & METHODS

### Study area

Due to the availability and completeness of raw data, this study only considered 13 major peanut-producing regions (Hebei, Liaoning, Jilin, Anhui, Fujian, Jiangxi, Shandong, Henan, Hubei, Hunan, Guangdong, Guangxi, and Sichuan), and 15 major rapeseed-producing regions (Inner Mongolia, Jiangsu, Zhejiang, Anhui, Jiangxi, Henan, Hubei, Hunan, Chongqing, Sichuan, Guizhou, Yunnan, Shaanxi, Gansu, and Qinghai) in China. The aforementioned peanut- and rapeseed-producing regions accounted for 86%–91% and 89%–98% of the national planting area from 2004 to 2023, respectively (*National Bureau of Statistics of China, 2025*). Additionally, they contributed 89%–93% and 94%–98% of the national production, respectively (*National Bureau of Statistics of China, 2025*). Due to the various natural conditions and cultivation levels, the planting area, yield, and production of peanut and rapeseed cultivation in different areas in China were different, as shown in Fig. 1. For peanut, Henan and Shandong had higher planting area, yield, and production, while Fujian, Jiangxi, and Hunan had lower planting area, yield, and production. In addition, Anhui had lower planting area and production, but its yield was the highest. For rapeseed,
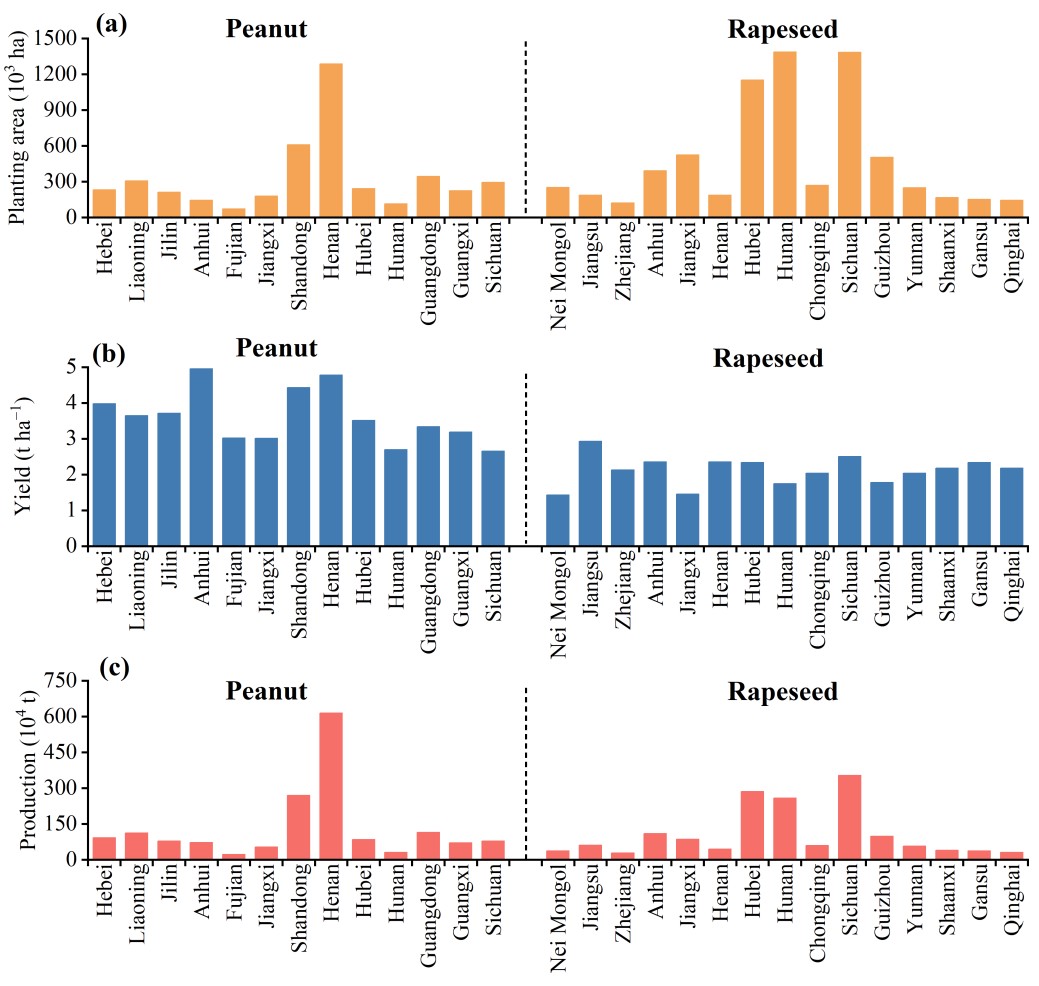

**Figure 1** Average planting area (A), yield (B), and production (C) of peanut and rapeseed in different regions during 2004–2023.

Hubei, Hunan, and Sichuan had higher planting area and production, while Zhejiang, Henan, Shaanxi, Gansu, and Qinghai had lower planting area and production. Jiangsu had low planting area and production, but it had higher yield. The specific geographic information of the study area was presented in Table S1.

## Data sources

Diverse farming practices result in varying CF and NF of crop production over different time series. A diachronic analysis can present the temporal dynamics characteristics of the CF and NF in peanut and rapeseed production. Limited to the availability of data, we obtained agricultural input data from 2004 to 2023, which included the amounts of fertilizers, agricultural film, and seeds from the National Agricultural Cost-Benefit Data (*NDRC, 2024*). The quantities of farm manure, pesticides, diesel used in agricultural machinery, and irrigation electricity were estimated according to the unit cost derived from the National Agricultural Cost-Benefit Data (*NDRC, 2024*) and the corresponding average

unit prices. The average unit prices of farm manure, pesticides, diesel, and electricity were obtained through peer-reviewed literature based on studies conducted in China, which were about 0.62 RMB kg$^{-1}$ (*Huang, Luo & Han, 2021*), 61.82 RMB kg$^{-1}$ (*Liu, 2020*), 6.00 RMB L$^{-1}$ (*NDRC, 2021*), and 0.50 RMB kwh$^{-1}$ (*NDRC, 2021*), respectively. Data on crop area and production were collected from the China Rural Statistical Yearbook (*National Bureau of Statistics of China, 2025*). Limited to the unavailable raw data of the related edible oil processing, energy consumption during the whole industrial chain of oil processing was derived from the national standard on the Norm of Energy Consumption Per Unit Product of Edible Vegetable Oil. According to the standards, the comprehensive energy consumption was set at 185 and 143 kgce t$^{-1}$ for edible oil production of peanut and rapeseed, respectively (*NDRC, 2013*).

## System boundary

This study adopted a cradle-to-factory gate LCA approach to quantify the CF and NF of edible peanut and rapeseed oils. As illustrated in Fig. 2, the system boundary includes raw materials and oil processing sections. The CF of the crop planting phase comprised upstream production of agricultural inputs (*i.e.,* fertilizers, pesticides, agricultural film, and seeds), N$_2$O emissions from in-field fertilization, carbon emissions from diesel-burning, and irrigation electricity. Similarly, the NF of the crop planting phase encompassed all direct and indirect Nr losses, including ammonia (NH$_3$) volatilization, emissions of nitrous oxide (N$_2$O) and nitric oxide (NO), nitrogen (N) leaching, and N runoff. For the oil processing section, the technological processes of edible peanut and rapeseed oil could be divided into pretreatment, squeezing, filtration, and refining. The oil extraction rate of peanut and rapeseed in China stabilized at around 35% and 34% in spatial–temporal dimensions, respectively, which came from the China National Grain and Oil Information Center. Therefore, it requires 2.86 t of peanut raw material and 2.94 t of rapeseed to produce 1 t of peanut and rapeseed oil, respectively.

## CF and NF of the crop planting system

Firstly, the accounting framework of carbon emissions and Nr loss from agricultural inputs was consistent with IPCC guidelines (*IPCC, 2019*), multiplying the activity data with corresponding emission factors as shown in the following equations:

$$CF_{input} = \sum(AD_i \times EF_{C_i}) \tag{1}$$

$$NF_{input} = \sum(AD_i \times EF_{Nr\_i}) \tag{2}$$

where $CF_{input}$ (kg CO$_2$eq ha$^{-1}$) and $NF_{input}$ (kg Nr ha$^{-1}$) are the carbon (C) emissions and Nr loss of agricultural inputs, respectively; $AD_i$ is the activity data of agricultural input $i$ (kg ha$^{-1}$ for fertilizer, pesticides, agri-film, and seeds, L ha$^{-1}$ for diesel fuel, kWh ha$^{-1}$ for irrigation electricity); $EF_{C\_i}$ is the carbon emission factor of the corresponding agricultural input $i$ (Table S2), and the $EF_C$ of irrigation electricity consumption is local and distinct in different regions of China (Table S3); $EF_{Nr\_i}$ is the Nr emission factors of the corresponding agricultural input $i$ (Table S4).

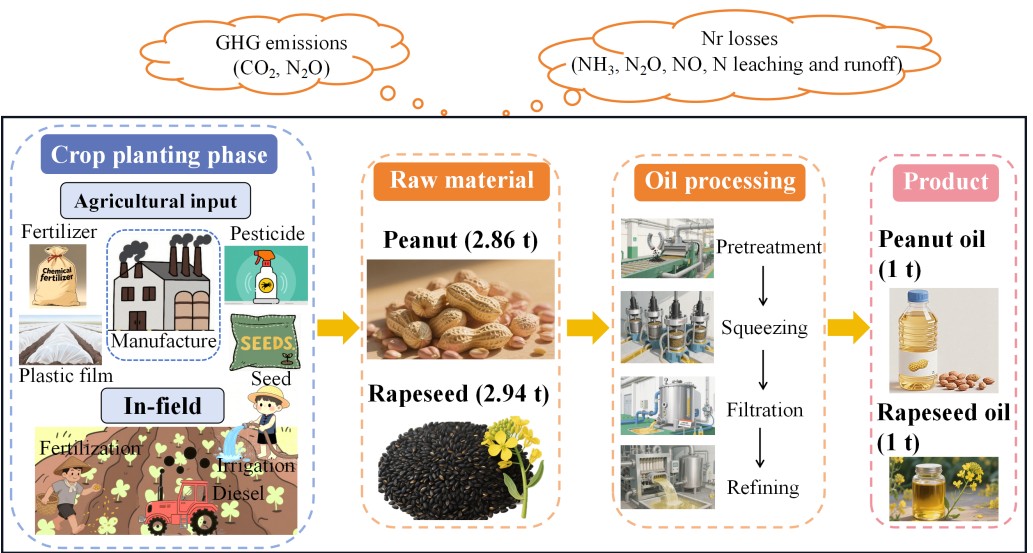

**Figure 2  System boundary definition of this study.**

Secondly, the direct and indirect $N_2O$ emissions associated with in-field N fertilizer application were calculated by the following equations:

$$CF_{direct} = \sum (F_n \times EF_1 \times (44/28) \times 273) \tag{3}$$

$$CF_{indirect} = \sum (F_n \times 11\% \times EF_2 \times (44/28) \times 273) + \sum (F_n \times 24\% \times EF_3 \times (44/28) \times 273) \tag{4}$$

where $CF_{direct}$ is the direct $N_2O$ emissions (kg $CO_2$eq ha$^{-1}$); $CF_{indirect}$ is the indirect $N_2O$ emissions (kg $CO_2$eq ha$^{-1}$); $F_n$ is the application rate of N fertilizers (kg ha$^{-1}$); 44/28 is the mass conversion factor of $N_2O$-N to $N_2O$; 273 is the global warming potential of $N_2O$ in a 100-year horizon (*IPCC, 2021*); 11% represents the fraction of atmospheric deposition of N volatilized from N fertilizer (*IPCC, 2019*); 24% represents the fraction of leaching and runoff from N fertilizer (*IPCC, 2019*); $EF_1$ represents the direct $N_2O$ emission factor from N inputs in different regions of China (Table S5); $EF_2$ and $EF_3$ represent the indirect $N_2O$ emission factor caused by N deposition (1%), and N leaching and runoff (0.75%), respectively (*IPCC, 2019*). These emission factors and fractions follow IPCC guidelines, which are widely used for large-scale estimates of farmland $N_2O$ emissions.

Simultaneously, the different forms of direct Nr loss in the field were calculated using the following equation:

$$NF_{direct} = \sum (F_n \times EF_j) \tag{5}$$

where, $NF_{direct}$ is the indirect Nr loss in the field (kg Nr ha$^{-1}$); $F_n$ is the application rate of N fertilizers (kg ha$^{-1}$); $EF_j$ is the emission factor of the Nr form $j$ (Table S6).

Finally, the CF and NF of the cropping system were calculated by the following equations:

$$CFa = (CF_{input} + CF_{direct} + CF_{indirect})/1000 \tag{6}$$

$$NFa = NF_{input} + NF_{direct} \tag{7}$$

where, $CFa$ (t $CO_2$eq ha$^{-1}$) and $NFa$ (kg Nr ha$^{-1}$) are CF and NF per unit area, respectively; 1,000 is the unit conversion factor.

The CF and NF by different functional units were calculated by the following equations:

$$CFt = (CFa \times A)/1000000 \tag{8}$$

$$CFy = CFa/Y \times 1000 \tag{9}$$

$$NFt = (NFa \times A)/1000000 \tag{10}$$

$$NFy = NFa/Y \times 1000 \tag{11}$$

where, $CFt$ (Tg $CO_2$eq) and $NFt$ (Gg Nr) are total CF and NF, respectively; carbon footprint per unit yield (CFy) (kg $CO_2$eq kg$^{-1}$) and nitrogen footprint per unit yield (NFy) (g Nr kg$^{-1}$) are CF and NF per unit yield, respectively; $A$ is the planting area (ha); $Y$ is crop yield per unit area (kg ha$^{-1}$); 100,0000 and 1,000 are both unit conversion factors.

## CF and NF of raw materials and oil processing

This study identified 1 t of edible peanut and rapeseed oil as the functional unit to compare the differences in CF and NF between peanut and rapeseed oil.

In the supply of raw materials, the CF and NF of peanut oil and rapeseed oil were estimated using the following equations:

$$CF_{raw\_mat} = CFy \times \sigma \times 1000 \tag{12}$$

$$NF_{raw\_mat} = NFy \times \sigma \tag{13}$$

where, $CF_{raw\_mat}$ (kg $CO_2$eq t$^{-1}$) and $NF_{raw\_mat}$ (kg Nr t$^{-1}$) are the CF and NF per unit of edible oil, respectively; $\sigma$ (t) is the raw materials required to produce 1 t of edible oil, estimated as 2.86 for peanut and 2.94 for rapeseed, respectively; 1,000 is the unit conversion factor.

Considering that the main driving force of oil processing is electric energy, comprehensive energy consumption is converted into electricity to estimate the CF and NF during the processing stage of peanut oil and rapeseed oil. The calculation formulas were established as follows:

$$CF_{oil\_pro} = P \times 8.167 \times 1.23 \tag{14}$$

$$NF_{oil\_pro} = P \times 8.167 \times 0.00329 \tag{15}$$

where, $CF_{oil\_pro}$ (kg $CO_2$eq t$^{-1}$) and $NF_{oil\_pro}$ (kg Nr t$^{-1}$) are the CF and NF per unit of edible oil, respectively; $P$ (kgce t$^{-1}$) is the comprehensive energy consumption per unit

edible oil; 8.167 is the conversion factor of energy consumption to electric power (kwh); 1.23 is the GHG emission factor of electric power (kg $CO_2$eq $kWh^{-1}$) (*Huang et al., 2017*); 0.00329 is the Nr emission factor of electric power (kg Nr $kWh^{-1}$) (*Chen et al., 2019*).

Therefore, the total CF and NF of the edible oil product were calculated using the following equations:

$$CF = CF_{raw\_mat} + CF_{oil\_pro} \qquad (16)$$

$$NF = NF_{raw\_mat} + NF_{oil\_pro} \qquad (17)$$

where, $CF$ (kg $CO_2$eq $t^{-1}$) and $NF$ (kg Nr $t^{-1}$) are the total CF and NF per unit of edible oil, respectively.

## Data processing and visualization

Data processing and visualization were performed using Microsoft Office Excel 2019 (Microsoft, Redmond, WA, USA) and Origin 2021. To explore the annual changing trends in CF and NF of the crop planting phase, the slope of linear regression at a *P*-value less than 0.05 for 2004–2023 was conducted with SPSS 22.0 (IBM Corp., Armonk, NY, USA). Therein, linear regression analyses for the total carbon footprint (CFt) and total nitrogen footprint (NFt) and nitrogen footprint per unit area (NFa) of rapeseed and NFy of peanut are not displayed because the *P*-value for the linear regression slope is greater than 0.05.

# RESULTS

## Temporal variation in CF and NF by functional unit

The CF and NF presented similar temporal change trends to some extent for both peanut and rapeseed. From 2004 to 2023, the CFt (Fig. 3A), CFy (Fig. 3C), NFt (Fig. 3D), and NFy (Fig. 3F) of peanut were consistently lower than those of rapeseed. Peanut and rapeseed had the lowest CFt and NFt in 2007, with 7.44 and 10.11 Tg $CO_2$eq of CFt, and 110.66 and 165.57 Gg Nr of NFt, respectively. For peanut, the CFt, carbon footprint per unit area (CFa) (Fig. 3B), NFt, and NFa (Fig. 3E) exhibited a significant increasing trend with an average annual growth value of 0.079 Tg $CO_2$eq, 0.010 t $CO_2$eq $ha^{-1}$, 2.150 Gg Nr, and 0.354 kg Nr $ha^{-1}$ during the research period, respectively. Meanwhile, the CFy of peanut showed a significant declining trend of 0.005 kg $CO_2$eq $kg^{-1}$ $yr^{-1}$ on average. For rapeseed, CFa, CFy, and NFy displayed a significant downward trend, with an average annual declining value of 0.010 t $CO_2$eq $ha^{-1}$, 0.015 kg $CO_2$eq $kg^{-1}$, and 0.185 g Nr $kg^{-1}$, respectively.

## Spatial variation in CF and NF by functional unit

The CF and NF were spatially heterogeneous across different regions in China (Fig. 4). Generally, regions with high CF also tended to have high NF discharge. For peanut, higher CFa (Fig. 4A) and NFa (Fig. 4B) mainly occurred in Hebei, Anhui, Jiangxi, Shandong, and Henan provinces, respectively with 2.12–2.56 t $CO_2$eq $ha^{-1}$ and 18.12–43.18 kg Nr $ha^{-1}$. There were larger peanut CFy (Fig. 4C) and NFy (Fig. 4D) apparent in Jiangxi and Sichuan, respectively with 0.62–0.74 kg $CO_2$eq $kg^{-1}$ and 10.01–12.35 g Nr $kg^{-1}$. Jiangxi had higher CF and NF than the other provinces, largely attributable to lower peanut productivity

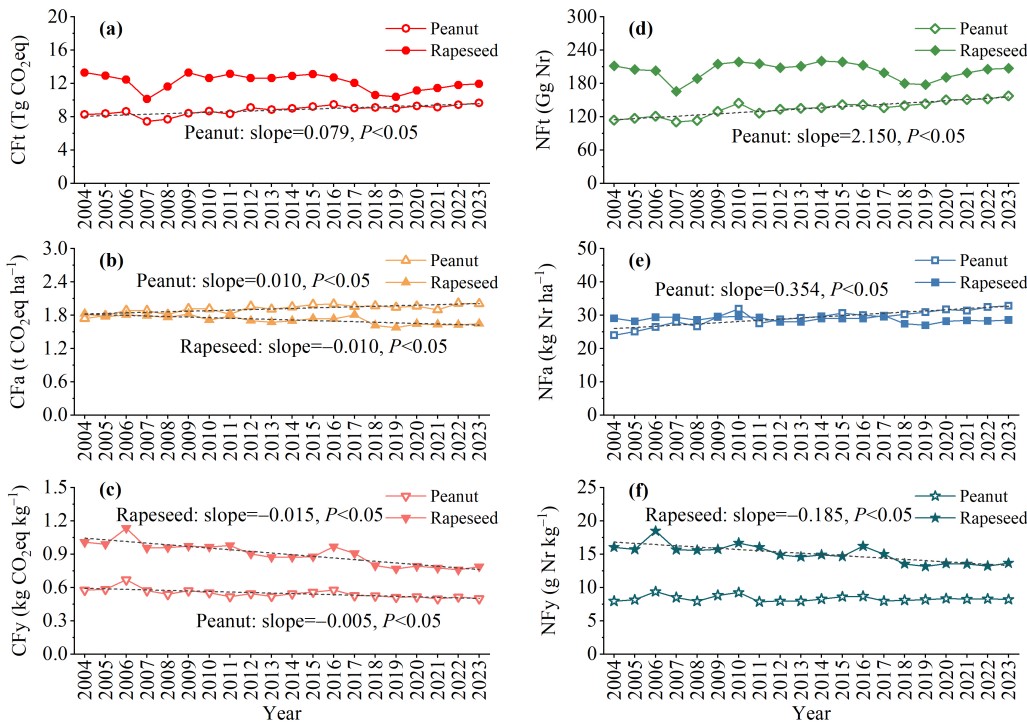

**Figure 3** **The CF (A, B, C) and NF (D, E, F) of peanut and rapeseed in China from 2004 to 2023.** CFt and NFt denote the total carbon and nitrogen footprint, respectively. CFa and NFa denote the carbon and nitrogen footprint per unit area, respectively. CFy and NFy denote the carbon and nitrogen footprint per unit yield, respectively. The black dotted lines indicate fitted linear regression lines.

or larger agricultural inputs. For rapeseed, higher CFa and NFa were mostly found in low-production provinces, including Jiangsu, Zhejiang, Anhui, Yunnan, Shaanxi, and Gansu, where they respectively ranged from 1.84 to 2.67 t $CO_2$eq ha$^{-1}$ and 35.45 to 41.72 kg Nr ha$^{-1}$. Higher rapeseed CFy and NFy were mainly distributed in Jiangxi, Hunan, Guizhou, Yunnan, Shaanxi, and Gansu, respectively with 0.85–1.13 kg $CO_2$eq kg$^{-1}$ and 16.03–17.80 g Nr kg$^{-1}$. In other words, rapeseed production in Yunnan, Shaanxi, and Gansu produced higher GHG emissions and also had higher Nr discharges.

## Contribution analysis of CF and NF for peanut and rapeseed

The most significant contributors to the CF and NF of peanut and rapeseed were fertilizer production and application. For the CF, N fertilizer and compound fertilizer emerged as the primary sources, respectively contributing 35.0% and 28.7% for peanut and 74.4% and 17.0% for rapeseed (Fig. 5A). The rapeseed CFy from N fertilizer was 3.5 times that of peanut (678.29 *vs.* 193.01 g $CO_2$eq kg$^{-1}$) (Fig. 5B). In addition, agricultural film and seed also played significant roles in the CF of peanut, while they contributed relatively minor to the CF of rapeseed. The peanut CFy from agricultural film and seed were 97.4 and 23.1 times those of rapeseed, respectively. Moreover, N runoff (30.4% for peanut and 34.0% for rapeseed) and NH$_3$ volatilization (39.0% for peanut and 33.9% for rapeseed) dominated the NF, followed by N leaching, NO emissions, and N$_2$O emissions (Fig. 5C).

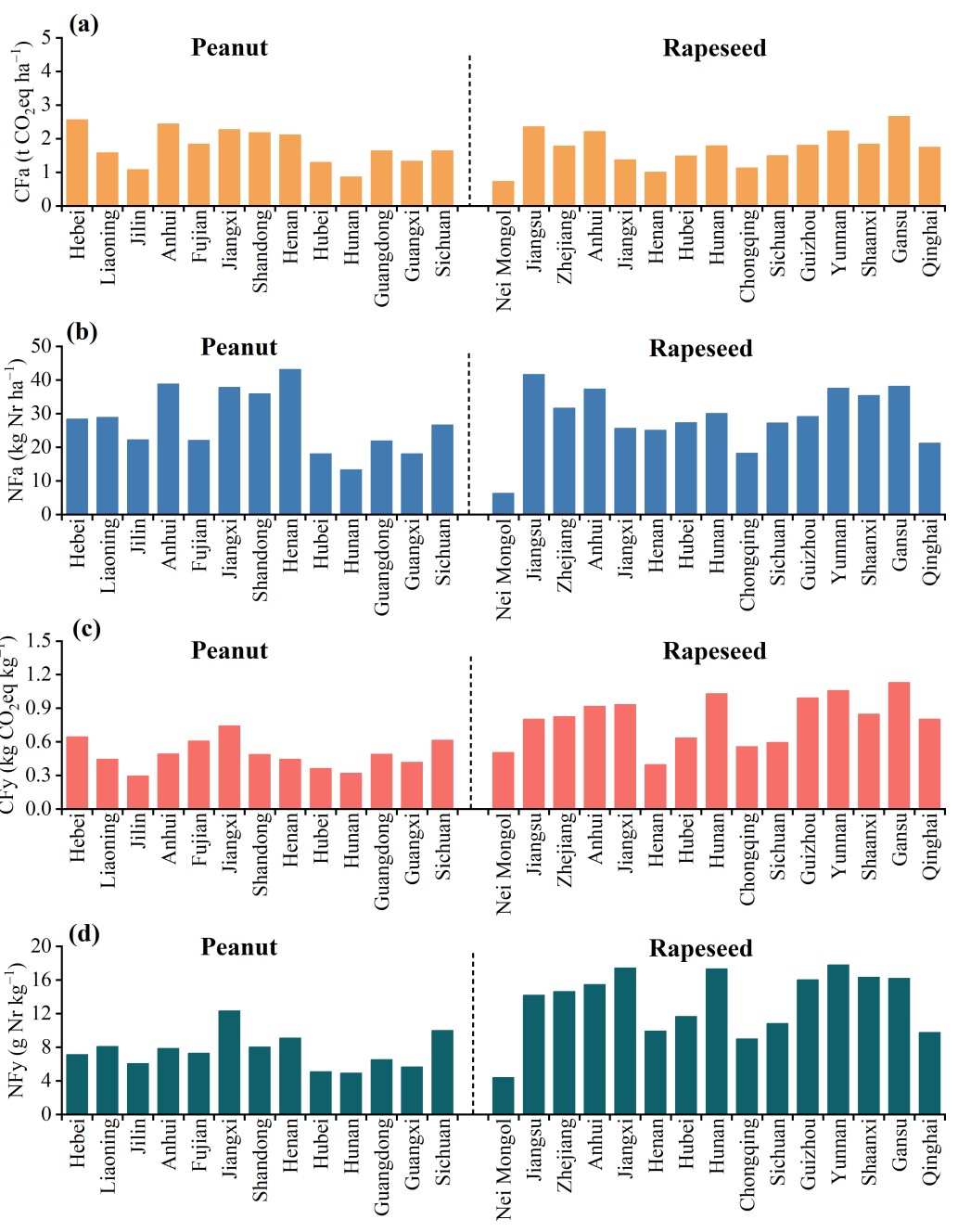

**Figure 4    Distribution of average CF (A, C) and NF (B, D) in the study region during 2004–2023.**

The rapeseed Nfy from N runoff and $NH_3$ volatilization were 1.8 and 1.4 times those of peanut, respectively (Fig. 5D).

## CF and NF of peanut oil and rapeseed oil

The CF of peanut and rapeseed oil, on average, were 3,312.2 and 3,722.4 kg $CO_2$eq t⁻¹ oil, respectively (Fig. 6). The corresponding NF were 28.5 and 43.4 kg Nr t⁻¹ oil for peanut

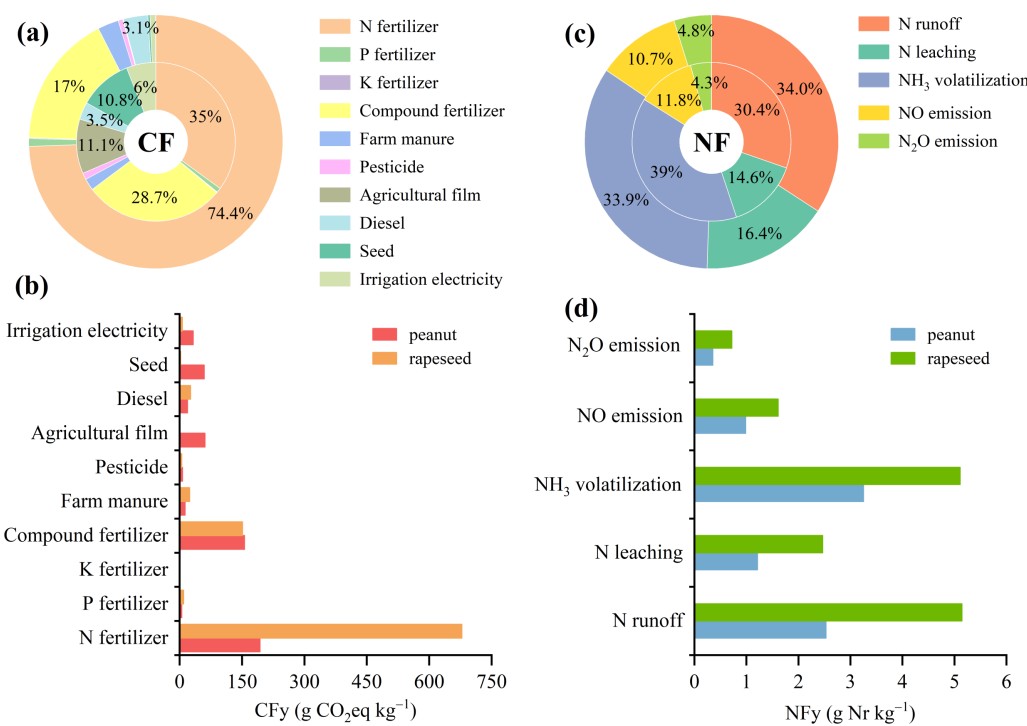

**Figure 5** Average contribution of various sources to the CF and NF of planting phase during 2004–2023. (A) Percentage contribution of different sources to the CF. (B) Percentage contribution of different sources to the NF. The inner circle represents peanut, and the outer circle represents rapeseed. (C) Contribution of different sources to the yield-scaled CF. (D) Contribution of different sources to the yield-scaled NF.

and rapeseed oil, respectively. The results illustrated that the CF and NF of peanut oil were lower by 11.0% and 34.2% than those of rapeseed oil, respectively. For the CF, raw materials contributed less than oil processing in peanut oil (43.9% *vs.* 56.1%), whereas the opposite occurred in rapeseed oil (61.4% *vs.* 38.6%). For the NF, almost all Nr losses were derived from raw materials, which comprised 82.6% and 91.1% in peanut and rapeseed oil, respectively. While the contributions of oil processing to the NF of peanut and rapeseed oil were both very low. Moreover, the CF and NF from raw materials in rapeseed oil were 1.6 and 1.7 times that of peanut oil, respectively. However, the CF and NF from oil processing in peanut oil were 1.3 times that of rapeseed oil.

## DISCUSSION

This study showed that both the CF (3,312.2 *vs.* 37,722.4 kg $CO_2$eq $t^{-1}$ oil) and NF (28.5 *vs.* 43.4 kg Nr $t^{-1}$ oil) of 1 t peanut oil were lower by 11.0% and 34.2% than those of 1 t rapeseed oil, respectively (Fig. 6). This significant discrepancy underscores the eco-friendliness of peanut oil relative to rapeseed oil in the pursuit of low-carbon and sustainable edible vegetable oils. Notably, the CF and NF of peanut oil processing were greater than those of rapeseed oil. The variances were attributed to the larger comprehensive energy consumption of peanut than that of rapeseed during the oil processing phase. The

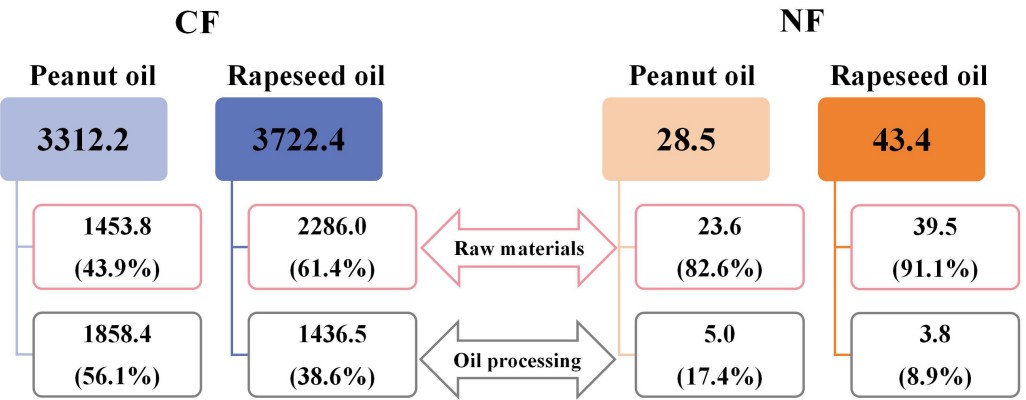

**Figure 6** **The CF (kg CO₂eq t−1 oil) and NF (kg Nr t−1 oil) of peanut and rapeseed oil.** The data was the average CF and NF during 2019–2023 because of the crop planting system's relatively flat CFy and NFy. The values in parentheses indicate the CF (NF) percentage from raw materials or oil processing to the total oil product CF (NF).

difference in energy consumption is attributed to the technological processes of the two edible vegetable oils. For example, during the pretreatment step, the oil-bearing seeds of peanut include cleaning, crushing, peeling, steamer, and frying, while the oil-bearing seeds of rapeseed include cleaning and frying (*Bai et al., 2021*). If high-energy efficiency technology and clean energy can be adopted, then peanut oil can demonstrate a greater advantage in low-carbon benefits. *Ji et al. (2021)* reported that if biofuel can be applied to replace diesel in the future, the CF of edible oil production can be reduced by 126 kg CO₂eq t$^{-1}$ oil (3.3%). When diesel is replaced with natural gas of the same calorific value, it can reduce the total carbon emissions by 7.9% in the edible oil refining sector (*Yang, Shi & Lu, 2023*). In addition, based on heat integration and process changes, such as exchange of multi-stage heat and installation of heat recovery devices, a 20% reduction in carbon emissions intensity can be achieved (*Ramanath et al., 2023*). The primary source of the CF and NF of oil products was raw materials, accounting for the main variation between 1 t peanut oil and 1 t rapeseed oil. Due to the comparable oil extraction rate, the disparities in the CF and NF of raw materials between peanut oil and rapeseed oil were primarily derived from the crop planting phase, consistent with the findings of *Bai et al. (2021)*. Peanut and rapeseed are both important oilseed crops, but they differ significantly in terms of biological characteristics, planting conditions, and management methods. In line with previous studies (*He et al., 2021*; *Ji et al., 2021*; *Li et al., 2022*), fertilizer production and application in farmland, particularly N fertilizer and compound fertilizer, contributed larger to the CF and NF in the cropping system (Figs. 5A, 5C). The CF from N fertilizer of rapeseed was much higher than that of peanut (Figs. 5A, 5B), attributed to the high N fertilizer input required to ensure rapeseed yield. Peanut was less dependent on N fertilizer than rapeseed, as the robust N₂ fixation of peanut root nodules could meet 40%–50% of its own N nutrient demand (*Gonzalez & Marketon, 2003*; *Wu et al., 2016*). Meanwhile, the N-use efficiency of peanut surpassed that of rapeseed (*Liu et al., 2023*). There was no obvious difference in the CFa (Fig. 3B) and NFa (Fig. 3E) between peanut and rapeseed,

while the CFy (Fig. 3C) and NFy (Fig. 3F) of peanut were consistently lower than that of rapeseed, on account of the higher peanut yield than that of rapeseed (Fig. 1). Additionally, peanut prefers warm conditions, and is generally cultivated with plastic mulch to improve heat conditions, while rapeseed has strong cold resistance (*Jian et al., 2020*; *Zhao et al., 2023*). Consequently, the use of agricultural film contributed a significant proportion to the CF of peanut, but its share was very low in the CF of rapeseed (Fig. 5A). Moreover, as the variances in planting density, seed morphology, and planting patterns between peanut and rapeseed, the amount of seeds per unit area of peanut is significantly higher than that of rapeseed, resulting in a large CF from seed in peanut (Figs. 5A, 5B). Therefore, in the context of environmental sustainability, this study advocates for an increased focus on peanut oil production and consumption.

Given that fertilizers, especially N fertilizer and compound fertilizer, contribute the most to the overall CF, optimizing fertilization practices and enhancing fertilizer efficiency are crucial strategies for reducing environmental emissions. In addition, $NH_3$ volatilization and N runoff were the main sources of NF (Fig. 5C), with rapeseed showing significantly higher losses than peanut (Fig. 5D). Other studies also reported that N fertilizer input was the primary source of NF, with $NH_3$ volatilization and N runoff being the dominant pathways (*Huang et al., 2023*; *Li et al., 2025*; *Li et al., 2022*). N fertilizer applications are known to promote the release of multiple Nr species, exhibiting linear or exponential response relationships (*Cui et al., 2013*). This underscores the critical need for targeted N management strategies, especially in rapeseed production systems. There have been mitigation policies on reducing key sources of GHG and Nr emissions from the cropping system. For example, China released the "National Soil Testing and Formula Fertilization Project" in 2005 and launched the "Zero-Growth of Chemical Fertilizer Use Action" in 2015. However, we found that the CFt (Fig. 3A), CFa (Fig. 3B), NFt (Fig. 3D), and NFa (Fig. 3E) of peanut showed a significant increasing trend from 2004 to 2023, which could be ascribed to higher agricultural inputs to obtain high production. Meanwhile, the increased yield during the period mainly affected the significant decrease in the CFy (Fig. 3C) and Nfy (Fig. 3F). These indicated that farming practices, especially chemical fertilizer in peanut and rapeseed cropping, still require improvement. Efforts to optimize fertilization practices should involve a judicious reduction in chemical fertilizer amounts and the necessary adjustment of fertilization structures. The study of *Bai et al. (2023)* showed that a 30% reduction in N fertilization resulted in a 35.1% reduction in CF and a 24.5% reduction in NF without compromising crop yields compared to conventional N fertilization. In a rapeseed cropping system, a balanced fertilization based on soil fertility led to a 15% reduction in N fertilizer application, decreased GHG emissions, and increased crop yield (*Li et al., 2019*). Moreover, the incorporation of organic materials such as biochar and biofertilizer, along with the use of controlled-release N fertilizer and urease or nitrification inhibitors, represented viable options to reduce GHG emissions and Nr loss while enhancing N use efficiency of crops (*Chen et al., 2021*; *Recio et al., 2020*; *Tan et al., 2018*; *Xia et al., 2017*). Field experiments have shown that both biochar alone and co-applied biochar with chemical fertilizer can increase peanut N uptake and yield while concurrently reducing $N_2O$ emissions and $NH_3$ volatilization compared to no application of biochar (*Agegnehu et al.,*

*2015*; *Tan et al., 2018*; *Wang et al., 2022*). Similarly, applying controlled-release compound fertilizer increased peanut yield by 8.9% and decreased cumulative emissions of $CO_2$ and $N_2O$ by 20.4%–45.4% compared to common compound fertilizer (*Liu et al., 2022*). *Shikha et al. (2023)* reported that applying biochar-based biofertilizer (rhizobium inoculants) increased peanut N uptake and reduced the need for N fertilizer while also lowering 6.6 kg $CO_2$eq ha$^{-1}$ GHG emissions by sequestering soil organic carbon stock. For rapeseed, combinations of urease and nitrification inhibitors have proven effective in reducing $NH_3$ volatilization and $N_2O$ emissions by more than 50% compared to chemical fertilizers alone (*Corrochano-Monsalve et al., 2021*). Moreover, systematic and linked measures have a stronger emission reduction effect than using them individually (*Chen et al., 2021*). A recent global meta-analysis by *You et al. (2024)* demonstrated that, adopting optimized management practices, including straw returning, biochar, optimized fertilization, cover cropping, and zero tillage so on, significantly mitigated Nr losses, with average reductions of 31% for $N_2O$, 23% for $NH_3$, 18% for N run-off, and 17% for N leaching. In summary, adopting these fertilization management practices is strongly recommended for improving Chinese peanut and rapeseed cropping systems.

Comparing the CF and NF of peanut and rapeseed in different regions, it was found that high CF and NF regions, like peanut planting in Hebei, Anhui, Jiangxi, Shandong, and Henan, and rapeseed planting in Jiangsu, Zhejiang, Anhui, Yunnan, Shaanxi, and Gansu (Fig. 4), belonged to the "hot spots" that should be paid attention to reduce the GHG and Nr emissions. The main driver was the high amount of N and compound fertilizer, which caused significant reactive Nr losses, especially in N runoff and $NH_3$ volatilization types (Fig. S1). Studies have indicated that replacing higher CF crops with lower CF crops could reduce GHG emissions from crop planting systems (*Wang et al., 2018*; *Zhang et al., 2017*). Therefore, based on reducing environmental costs, it would be better not to crop peanut in Jiangxi and rapeseed in Yunnan, Shaanxi, and Gansu, where they have high CF and NF but low productivity. In relative to rapeseed, peanut develop below the ground and have specific soil temperature and moisture requirements. Agricultural film mulching offers key advantages in maintaining soil temperature and saving water, thus improving crop production (*Berger et al., 2013*). The results showed that agricultural film contributed significantly to the CF of peanut, but it contributed very low to the CF of rapeseed (Fig. 5A). In peanut planting regions, Shandong had the most extensive application of film mulching, reflected in the considerable contribution to the CF, followed by low production areas Hebei, Anhui, and Hubei (Fig. S1). *Chen et al. (2023)* indicated that replacing polyethylene film mulch with biodegradable plastic film mulch not only could control the plastic residue pollution, but also induced lower GHG and Nr emissions intensity than a no-plastic mulching cultivation system. Peanut planting areas with low rainfall often coincide with high temperatures and drought, necessitating more irrigation and associated electricity consumption. Due to the low rainfall in Hebei and Henan, the contribution of irrigation electricity to the CF was higher than that in other regions (Fig. S1). *Wang et al. (2022)* showed that drip irrigation, especially mulching drip irrigation, not only could increase peanut yield but also induced less $NH_3$ and $N_2O$ emissions, as compared to furrow irrigation. Moreover, diesel combustion of agricultural machinery also led to

comparable CF of peanut and rapeseed. With the ongoing agricultural modernization in China, the agricultural mechanization level is expected to improve further (*China Government Network, 2016*). These results emphasized the critical need to adopt advanced film mulching technology, water-saving irrigation technology, and cleaner renewable energy that judiciously reduces GHG and Nr emissions in agriculture.

This study quantified the CF and NF of peanut oil and rapeseed oil. Meanwhile, there are some uncertainties and limitations from different sources. First, the uncertainty stems from the choice of emission factors. For example, we only used default emission factors of NO emission, $NH_3$ volatilization, and N runoff/leaching from N fertilizer, which may introduce uncertainty to the NF of different provinces. It should be noted that variations in regional climate, crops, and soil types can lead to discrepancies in emission factors, which require further investigation. Second, due to data limitations, the comprehensive energy consumption of edible oil processing is based on the national industry standard that is not differentiated from regions and years; thus, this study does not consider spatial and temporal variations in the CF and NF of edible oil processing. Third, this study does not consider transportation and packaging stages in the CF and NF calculation. Although this study utilized national statistical data and peer-reviewed sources, which offer consistency across spatial and temporal scales, it is acknowledged that reliance on secondary data may limit the precision of footprint estimates. Future research should incorporate field-level data collection and region-specific emission factors to show more detailed information on the CF and NF of edible peanut and rapeseed oil production in China.

## CONCLUSIONS

This study employed a cradle-to-factory gate life cycle assessment method to quantitatively evaluate peanut and rapeseed oil's carbon and nitrogen footprint in China. The CF and NF of peanut oil were 3,312.2 kg $CO_2$eq $t^{-1}$ oil and 28.5 kg Nr $t^{-1}$ oil, respectively, compared to 3,722.4 kg $CO_2$eq $t^{-1}$ oil and 43.3 kg Nr $t^{-1}$ oil for rapeseed oil. This demonstrated that peanut oil had better environmental performance than rapeseed oil from a CF and NF perspective in China, representing 11.0% reduction in CF and 34.2% reduction in NF for peanut oil relative to rapeseed oil. The primary divergence in environmental impacts stems from the cropping phase, where peanut cultivation consistently demonstrates lower yield-based CF and NF than rapeseed during 2004-2023. Fertilizer application, particularly N and compound fertilizers, was the dominant contributor, meanwhile N runoff and $NH_3$ volatilization were the major sources of NF. Peanut cropping in Jiangxi and rapeseed cropping in Yunnan, Shaanxi, and Gansu, showed higher CF and NF while presenting lower productivity, implying that optimizing peanut and rapeseed layout in China would be of significant importance to developing the sustainable industry of edible vegetable oil. This study provides critical insights for policymakers and stakeholders aiming to achieve low-carbon, sustainable agricultural practices in China's edible oil sector.

### Funding

This work was supported by the Agricultural Science and Technology Innovation Program of China (No. CAASZDRW202417 and No. CAAS-CSGLCA-202301) and the National Natural Science Foundation of China (No. D41105115). The funders had no role in study design, data collection and analysis, decision to publish, or preparation of the manuscript.

### Grant Disclosures

The following grant information was disclosed by the authors:
Agricultural Science and Technology Innovation Program of China: No. CAASZ-DRW202417 and No. CAAS-CSGLCA-202301.
National Natural Science Foundation of China: No. D41105115.

### Competing Interests

The authors declare there are no competing interests.

### Author Contributions

- Fen Ma conceived and designed the experiments, performed the experiments, analyzed the data, prepared figures and/or tables, authored or reviewed drafts of the article, and approved the final draft.
- Mingbao He conceived and designed the experiments, performed the experiments, prepared figures and/or tables, and approved the final draft.
- Yingchun Li conceived and designed the experiments, analyzed the data, authored or reviewed drafts of the article, and approved the final draft.
- Yanqun Wang analyzed the data, authored or reviewed drafts of the article, and approved the final draft.
- Zhengping Peng analyzed the data, authored or reviewed drafts of the article, and approved the final draft.
- Yinlong Xu analyzed the data, authored or reviewed drafts of the article, and approved the final draft.
- Bohan Zhao analyzed the data, prepared figures and/or tables, and approved the final draft.
- Jingyu Zhang analyzed the data, prepared figures and/or tables, and approved the final draft.

### Data Availability

 Raw data is available in the Supplemental Files.

### Supplemental Information

Supplemental information for this article can be found online at http://dx.doi.org/10.7717/peerj.19941#supplemental-information.

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
