# Peer review of "Peanut oil is more environmentally sustainable than rapeseed oil from a carbon and nitrogen footprint perspective in China"

_PeerJ, doi:10.7717/peerj.19941_

## Round 0.1 · original submission · Major Revisions

· Academic Editor

Major Revisions

Carbon and nitrogen footprint emissions are closely linked to climate change. Therefore, your study offers valuable insights for oilseed growers and decision-makers. However, certain technical aspects need to be addressed to improve the clarity of your article. I strongly recommend carefully reviewing the reviewers' suggestions and thoughtfully considering each recommendation. If you disagree with any suggestion, providing clear and well-reasoned justifications would be beneficial. Additionally, your article requires linguistic refinement. You might consider seeking assistance from a colleague with expertise in scientific editing or utilizing our professional editing service to achieve the highest standards of clarity and readability.

Reviewer 1 ·

Basic reporting

Oilseeds play a crucial role in the agricultural production system. In the current context of climate change, understanding their emission potential is essential for optimizing production and processing activities. In this regard, the study on quantifying the emission potential of peanuts and rapeseed is highly relevant and timely. As highlighted by the author through their review of literature, many studies have focused on carbon and nitrogen footprinting separately concerning the production and processing of these crops. Identifying this gap, the authors have conceptualized a comprehensive study that jointly analyzes both production and processing activities for carbon and nitrogen footprints in peanuts and rapeseed. This integrated approach is commendable.

Experimental design

1 There is no need to restate introductory information about crops in the Materials and Methods section. For instance, statements like “For edible peanut oil and rapeseed oil production, peanut and rapeseed respectively are the raw materials” are redundant and should be omitted.
2 Replace phrases like “For edible peanut oil and rapeseed oil production” with “Edible oil production of peanut and rapeseed” to improve fluency and readability wherever such phrases appear.
3 Long sentences, such as “Peanut cultivation is mainly distributed in 13 regions, namely Hebei, Liaoning, Jilin, Anhui, Fujian, Jiangxi, Shandong, Henan, Hubei, Hunan, Guangdong, Guangxi, and Sichuan, where the planting area accounted for 86%–91% of the total planting area and the yield accounted for 89%–93% of the gross yield on average during 2004–2022 (NBS 2005–2023)”, hinder readability. Break these into shorter sentences or simplify their structure for better flow.
4 The description of the study site appears vague. Instead, provide specific geographic information, such as the region's name, latitude, longitude, and altitude. This will add clarity and precision.
5 There are significant concerns about the reliability of the results, as the data are solely derived from secondary sources like the National Agricultural Cost-Benefit Data, China Rural Statistical Yearbook, and peer-reviewed literature. It is advisable to include some primary data collection, such as representative samples from the study area, to provide stronger validation of the findings.
6 Statements like “The raw materials for peanut oil and rapeseed oil production are peanut and rapeseed, respectively, for which the cultivation pattern is similar” should be avoided throughout the manuscript. These are self-evident and detract from the article's overall credibility.
7 The methodology for calculating carbon footprint (CF) and nitrogen footprint (NF) is well-presented and clear.

Validity of the findings

Result
1 Correct Statement Grammar: The phrase “peanut were consistently lower than those of rapeseed” should be revised to “peanut were consistently lower than that of rapeseed” for grammatical accuracy.
2 Significant Findings: The authors have clearly presented the significant findings of the study, which is commendable.
3 Enhance Results Section: Including additional insights into carbon footprint (CF) and nitrogen footprint (NF) in the Results chapter would significantly enrich the article and provide greater depth to the discussion.
Discussion
1 Lack of Ground Truth Data: As mentioned earlier, the study suffers from the absence of ground-truth data. If the research had partially incorporated primary data collection, it would have provided more robust insights into oilseed production and its impact on emissions. This inclusion could have strengthened the study’s reliability and relevance.
2 Discussion Section Deficiencies: In the discussion, the variations in emissions were primarily correlated and justified based on fertilization, irrigation water, and electricity use. However, canola and peanut are fundamentally distinct crops, differing significantly in their growing period, sowing methods, intercultural practices, mechanization, harvesting, and processing. The current discussion lacks a comprehensive analysis that considers these contrasting production and processing practices. A more detailed comparison would have greatly enhanced the discussion's depth and rigor.
Conclusion
The conclusion is superficial; some more key findings may be brought into the conclusion part of the MS.

·

Basic reporting

In this study, the researchers aimed to determine which oil—peanut oil or canola oil—has a more advantageous environmental sustainability profile by comparing their carbon (CF) and nitrogen (NF) footprints using a "cradle-to-factory gate" life cycle assessment (LCA) method. I suggest that the researchers carefully address the following points:

• The summary section of the article should be better supported by numerical data. This will likely generate more interest in the researchers' article.

• The introduction of the article relies on specific main sources regarding carbon and nitrogen footprints. However, these sources are not recent; they are outdated and do not qualify as academically respectable or validated publications. Therefore, the sources provided in the introduction need to be revisited. For example, there are sources from 2013 or earlier years. Many of these sources currently conflict with new and up-to-date data.

• Additionally, it seems that the researchers' interpretations based on reports may not effectively result in accurate data acquisition. I believe this aspect also needs to be re-evaluated.

• In the study, the researchers have focused more on carbon footprints and mentioned only a limited number of studies on nitrogen footprints. For instance, the types of nitrogen losses (e.g., NH3 volatilization, N2O emissions) and their management strategies have been studied less thoroughly.

• It is unclear how regional differences are addressed in the literature. For example, the impact of different soil types and climates on emission factors could be examined more comprehensively.

Experimental design

- The connection of the selected species to China should be emphasized more. This way, readers can better understand why these plants were chosen.
- Researchers also state that they conducted CF and NF calculations using the "cradle-to-factory gate" life cycle assessment (LCA) method. The fact that areas such as transportation, packaging, and final consumption, which may have been overlooked in this calculation method, were not addressed may not be adequate for perceiving the article's environmental impact assessment.
- The emission factors used and the calculation framework are solely based on IPCC guidelines. Regional and product-specific variables (for example, the effects of different soil types and climates) have not been sufficiently considered. This limits the examination of the results and their contributions.
- I recommend that the graphs presenting the results be clearer and more explanatory. They should also be free from different colors and shapes.

Validity of the findings

• I believe that particularly for researchers, expanding the methodological scope, diversifying data sources, and addressing regional differences in more detail will help prove the reliability of the study, and this situation should be applied to the entire article.

• While researchers indicate that the environmental impact of peanut oil is lower, it is also stated that peanut oil consumes more energy during the processing stage and results in higher carbon emissions. An analysis should be conducted that includes a look at the total energy consumption data in this situation, and there should also be a direction towards alternative energy sources.

• Recommendations for reducing fertilizer use have been limited, and the effects of biological fertilizers or technologies that increase nitrogen use efficiency have not been sufficiently discussed.

• The results should be highlighted more by referencing the literature more extensively, emphasizing the similarities or differences with previously conducted studies.

---

## Round 0.2 · accepted · Accept

· Academic Editor

Accept

I would like to thank you for accepting the referees' suggestions and improving your article based on their suggestions. Your article is ready to publish. We look forward to your next article.

·

Basic reporting

Thank you to the authors for making the corrections I indicated. The article is suitable for publication in its current form.

Experimental design

Thank you to the authors for making the corrections I indicated. The article is suitable for publication in its current form.

Validity of the findings

Thank you to the authors for making the corrections I indicated. The article is suitable for publication in its current form.

Additional comments

Thank you to the authors for making the corrections I indicated. The article is suitable for publication in its current form.